# Effect of Fly Ash and Metakaolin on Properties and Microstructure of Magnesium Oxysulfate Cement

**DOI:** 10.3390/ma15041334

**Published:** 2022-02-11

**Authors:** Tong Liu, Chunqing Li, Li Li, Wenqiang Fan, Yudong Dong, Huihui Liang, Hongjian Yang

**Affiliations:** 1School of Chemical Engineering and Technology, Hebei University of Technology, Tianjin 300130, China; 201921503015@stu.hebut.edu.cn (T.L.); dongyd2022@163.com (Y.D.); 202031504010@stu.hebut.edu.cn (H.L.); 2Tianjin Cheng An Thermal Power Co., Ltd., Tianjin 300161, China; l13820972973@163.com (C.L.); lili200702@126.com (L.L.); fanwq123@126.com (W.F.)

**Keywords:** MOSC, metakaolin, mechanical properties, water resistance, microstructure

## Abstract

To improve the mechanical performance and lower the production cost of magnesium oxysulfate cement (MOSC), this article investigates the effects of single and compounded addition of metakaolin (MK) and/or fly ash (FA) on the setting time, mechanical strength, water resistance, hydration product, composition, and microstructure of the resulting cement. MOSC samples with different proportions, ranging from 0 to 30 wt.%, of FA and/or MK substituting magnesium oxide (MgO) were prepared. The microstructure was explored by scanning electron microscopy, X-ray diffraction, and mercury intrusion porosimetry. The findings suggest that adding FA can delay the setting of MOSC; however, the effect of adding MK to MOSC was reversed. Furthermore, the phase composition of the MOSC hydration products was unaltered upon adding FA and/or MK, but thicker and longer 517 phase crystals were observed. FA and MK can effectively fill the large pores of MOSC through filling and nucleation effects, reduce the pore size, and form a denser microstructure, thereby improving its mechanical properties. The optimal MOSC sample was found by substituting 10 wt.% of both FA and MK, resulting in a cement that exhibited a short setting time and an incredibly high mechanical strength and density. These findings will further the development of stronger, more cost-efficient, and more water-resistant MOSC products.

## 1. Introduction

Magnesium oxysulfide cement (MOSC) is an environmentally benign and air-hardening, ternary system that is composed of MgO–MgSO_4_–H_2_O [1,2,3]. Compared to traditional Portland cement, MOSC is advantageous due to its faster setting, lighter weight, and greater mechanical strength [4]. Compared to magnesium oxychloride cement (MOC), MOSC does not absorb moisture or return halogens, and it exhibits low corrosion toward steel bars [5]. MOSC is widely used in building structures, walls, and decorative materials, as well as refractory products, and it exhibits excellent performance in its applications [6]. However, the cost of manufacturing MOSC is significantly higher than that of manufacturing traditional cement, which limits the necessary development of MOSC. MOSC has great potential if the cost can be reduced without compromising its stability and properties.

According to a previous study, the content of 3Mg(OH)_2_·MgSO_4_·8H_2_O (318 phase) does not exceed 50% at room temperature, which leads to the low compressive strength of MOSC [7]. In recent years, to improve the mechanical performance and water resistance properties of MOSC, many researchers directed their attention to incorporating various chemical additives into the system. These additives include citric acid and citrate [8], tartaric acid [9], sodium malate [10], and phosphoric acid and phosphate [11]. The addition of additives into the MOSC induces the formation of a stronger phase (517). Changing the phase composition of hydration products is recommended to improve the mechanical properties of cement-based materials.

The phase composition and microstructure of hydration products significantly affect the mechanical properties of the MOSC. In addition to changing the phase composition of hydration products via chemical additives, mineral admixtures can be added to MOSC to improve the microstructure of cement-based materials. Fly ash (FA) is a common type of cement-based filler, which is widely used in Portland cement (PC). When FA is added to PC, the cement pore size can be reduced by physical filling and pozzolanic effects, forming a more compact structure and, thus, improving the mechanical properties [12]. Furthermore, adding FA can reduce early hydration heat, mitigate micro-cracks, and improve the overall durability of concrete [13]. Since FA is a solid, combustion byproduct collected from coal-fired power plants, adding FA to traditional cements can improve the performance of the cement, reduce the environmental impact, and provide economic benefits. As an example, adding FA to MOC enhanced the workability of cement paste and slightly reduced the compressive strength, but significantly improved the water resistance [14]. In another instance where FA was added to MOSC, spherical FA particles improved the slurry uniformity and promoted the formation of the 517 phase in the hydration products via the nucleation effect [15].

Metakaolin (MK) is a mineral admixture that has drawn the focus of many scientists, but it remains rarely used in magnesium-based cement. However, due to its active aluminosilicate composition, MK is widely used in Portland cement [16]. When MK is added into Portland cement, the active component reacts with the hydration product Ca(OH)_2_ and further accelerates the hydration process [17]. Furthermore, adding MK can significantly improve the mechanical properties and freeze–thaw durability of traditional Portland cement [18]. Incorporating MK into MOC had been reported in terms of macro-performance such as a quantitative analysis of changes in quality and compactness and mechanical properties, as well as in terms of microstructure, such as an analysis of the phase composition of hydration products and microscopic morphology [19]. Integrating MK into magnesium phosphate cement (MPC) improved the 28 day compressive and tensile bonding strength of cement mortar, as well as the frost resistance and water resistance, while it reduced the drying shrinkage [20]. The effect of MK on the properties of MOSC has never been considered. Accordingly, this study adopts two different procedures, i.e., single and compound addition, to explore the synergistic effects of MK and FA on the properties of MOSC. The results of this study can serve as a baseline for addressing the economic challenges associated with magnesium cement brought upon by the recent price increase in its precursor material, MgO. In addition, these strategies enable an alternative route for enhancing the MOSC system, so as to further the advancement of the cements.

## 2. Materials and Methods

### 2.1. Raw Materials

The purities of light-burned magnesium powder (Huafeng Magnesium Mineral Products Co., Ltd., Haicheng, China) and MgSO_4_·7H_2_O (Yongxing Chemical Factory, Tianjin, China) were 85% and 99.0%, respectively. The activity of the light-burned powder was 62.0% as measured by the hydration method [21]. In each set of experiments, the content of citric acid was fixed at 0.5 wt.% (mass fraction of MgO, Kemiou Chemical Reagent Co., Ltd., Tianjin, China). The water used in this study was distilled water. The chemical compositions of the MgO, grade IFA (Yandong Mineral Products Co., Ltd., Tianjin, China) and MK (Hengyue Mineral Products Co., Ltd., Taiyuan, China) are summarized in Table 1. The physical properties of the raw materials are shown in Figure 1. The chemical composition and particle size distribution were measured by X-ray fluorescence (XRF, ARL QUANT X, ThermoFisher Scientific, Shanghai, China) and laser particle size analysis (LSPA, Anton Paar LitesizerTM500, Malvern, Worcs, UK). According to Figure 1a, the median particle sizes (D_50_) of MgO, MK, and FA were 30.29, 3.04, and 8.96 μm, respectively. Furthermore, the Brunauer–Emmett–Teller (BET) surface areas of MgO, FA, and MK were 1.917, 22.963, and 229.191 m^2^/g, respectively (Figure 1b).

### 2.2. Specimen Preparation

The samples without mineral admixtures were set aside as control group C, while the samples with solely MK or FA or with both MK and FA were denoted as the experimental groups C-MK, C-FA, and C-(MK+FA), respectively. The amount of magnesium powder replaced with MK or FA was set to 0%, 5%, 10%, 15%, 20%, 25%, and 30% (mass fraction of magnesium powder). The mass ratio of MK:FA was set to 1:1, when added in combination with MOSC. The specific mixing ratio of magnesium oxysulfide cement slurry is shown in Table 2. The total mass of raw materials was 1000 g.

MgSO_4_·7H_2_O was dissolved in water to prepare the magnesium sulfate solution at the required specific gravity (1.25 g/cm^3^). According to the ratio of α-MgO, MgSO_4_, and H_2_O, each raw material was calculated and accurately measured. Specific amounts of citric acid and FA or MK were mixed and stirred for 15 min. Then, the slurry was poured into a steel mold and shaken on a cement mortar shaker for 60 s. The size of the steel molds was 40 × 40 × 40 mm^3^ and 10 × 10 × 60 mm^3^. Finally, the steel mold containing the cement slurry was placed in the cement maintainer (standard curing procedure: 25 °C and 65% relative humidity) for 24 h and subsequently demolded, yielding the MOSC samples. Afterward, the MOSC samples were kept in a curing box for 28 days for performance evaluation. Three samples were selected from each group for performance testing, and the average value was taken as the final test result.

### 2.3. Test Methods

#### 2.3.1. Fluidity and Setting Time Tests of MOSC

The fluidity and setting time of the MOSC paste were evaluated according to Chinese National Standards GB/T8077-2012 and GB/T1346-2011, respectively. The initial setting time was set to 4 ± 1 mm on the Vicat apparatus. The final setting time was assigned to when the depth of the probe on the sample surface receded to less than 0.5 mm.

#### 2.3.2. Strength Test of MOSC

Mechanical properties of MOSC samples were assessed using an electron universal testing machine (CMT6104, Sans, Shenzhen, China) with a maximum load of 300 kN. The flexural and compressive strengths were tested according to the Chinese National Standard GB/T17671-1999, corresponding to loading speeds of 50 and 2400 kN/s. The size of each MOSC specimen used to calculate the softening coefficient was 40 × 40 × 40 mm^3^. Half of the prepared samples were tested for compressive strength after 28 days of curing, while the remaining samples were tested for immersion strength after being kept in water for predetermined durations. The water resistance, which can be expressed as the softening coefficient (R_f_), was calculated using Equation (1).
(1)$$R_{f}=\frac{R_{\left(w,n\right)}}{R_{\left(a,28\right)}},$$
where R_(w,n)_ is the compressive strength of the sample soaked in water for n days, and R_(a,28)_ is the compressive strength of the sample cured for the standard amount of 28 days.

#### 2.3.3. Phase Composition and Microstructure Analysis

The 28 day cured MOSC samples were crushed into slag, and then ground into powder. The powder was analyzed by XRD (XRD, D8 discover, Bruker AXS, Karlsruhe, Germany) with the following conditions: scanning step of 0.019° and scanning rate of 0.2 s/step. The sample components were quantitatively analyzed by the Rietveld method using Topas4.2 [22,23]. The cross-section of the sample was taken, and the microstructure was analyzed after gold spraying. The working voltage of the field-emission scanning electron microscope (Nova Nano FESEM450, FEI company, Hillsboro, OR, USA) was 10.00 kV. The pore size distribution of the treated debris of MOSC was analyzed by mercury intrusion porosimetry (MIP, Quantachrome Autoscan 60, Boynton Beach, FL, USA).

## 3. Results

### 3.1. Fluidity and Setting Time

C, C-FA, C-MK, and C-(MK+FA) indicate that the MOSC paste did not contain additives, whereby FA was added alone, MK was added alone, or MK and FA were mixed together.

Figure 2 shows the fluidity of C, C-FA, C-MK, and C-(MK+FA). Specifically, the fluidity of the C slurry was 144 mm. When the FA content was 30%, the fluidity of the C-FA slurry was 175 mm, a 21.53% increase. The fluidity of C-FA gradually increased as the FA content increased, confirming that the fluidity of MOSC paste can be improved with the addition of FA. Furthermore, when the content of MK was 30%, the fluidity of C-MK was 100 mm, an obvious 30.56% decrease. The fluidity of C-MK gradually decreased as MK content increased, confirming that the addition of MK reduces the fluidity of MOSC paste. When both MK and FA were added, the fluidity of the resulting C-(MK+FA) improved, increasing with FA and MK content. When the dosage of MK and FA was 30%, the fluidity of C-(MK+FA) was 11.11% higher than that of C. 

Figure 3 shows the setting profiles of C, C-FA, C-MK, and C-(MK+FA). According to Figure 3a, the setting time of C-FA was longer than that of C, and the setting time of C-FA gradually lengthened as the amount of FA increased. According to Figure 3b, when MK was added to MOSC, the setting time was reduced compared to that of C, further decreasing with the addition of MK, confirming that MK can quicken the setting procedure of C-MK. Figure 3c demonstrates the shorter setting time of C-(MK+FA) in contrast to sample C. As the MK and FA content (weight ratio 1:1) increased, the setting time of C-(MK+FA) gradually shortened, indicating that the compounded addition of MK and FA can hasten the setting process of C-(MK+FA). According to Figure 3, C-FA expressed the longest setting time at each admixture substitution level, followed by C-(MK+FA) and C-MK. 

### 3.2. Mechanical Properties

The mechanical properties of hardened MOSC can be quantified by its compressive and flexural strength. The compressive strength of MOSC samples cured at differing durations is summarized in Figure 4. Figure 4a shows the early compressive strength of MOSC after curing for 3 days. When the admixture component was under 10%, the early compressive strengths of C-MK, C-FA, and C-(MK+FA) all increased with the increase in admixture content. When the admixture content exceeded 10%, the compressive strength of the MOSC sample gradually decreased. When the dosage was 10%, the compressive strengths of C, C-FA, C-MK, and C-(MK+FA) samples reached the maximum values of 65.17, 67.62, 68.47, and 69.73 MPa, respectively. At the same dosage level, the compressive strength of C-(MK+FA) was higher than that of C-MK and C-FA, and the compressive strength of the three experimental groups was higher than that of C. In addition, the maximum strength of the specimen cured to 3 days merely reached 70% of that cured to 28 days. The trend detected in the compressive strength with respect to the admixture content after 28 days of curing was similar to that after 3 days of curing. Figure 4b shows that, after 28 days of curing, the compressive strength of hardened MOSC samples, whether single-doped or compounded with FA and MK, initially increased and then gradually decreased. C-(MK+FA) samples at 10% FA and MK exhibited the maximum compressive strength, followed by C-MK and C-FA in similar conditions; the corresponding compressive strength values were 105.70, 100.76, and 96.73 MPa, respectively. It can be seen that the addition of MK or FA had a positive effect on the mechanical properties of MOSC. The improvement of mechanical properties by the compound admixture of MK and FA was better than that by MK or FA alone. 

Figure 5 shows the flexural strength of MOSC samples cured for different durations. The general trend observed in the flexural strength was identical to that observed in the compressive strength (Figure 4). As the MK and FA portion increased, the flexural strength initially increased, and then decreased. At 10% admixture content, the C-(MK+FA) sample exhibited the greatest maximum flexural strength, followed by C-MK and C-FA, at 29.82 MPa, 28.62 Mpa, and 25.42 MPa; in contrast to sample C, the flexural strength increased by 48.36%, 42.32%, and 26.41%, respectively. The improvement effect of FA and MK on the flexural strength of the system was better than that on the compressive strength. Figure 6 contrasts the bending–compressive strength ratio of each MOSC sample. When the admixture content was set to 30%, the maximum bending–compressive strength ratios of C-FA, C-MK, and C-(MK+FA) were observed at 0.26, 0.28, and 0.30, respectively. The results showed that MK, FA, and compounded MK and FA positively affected the flexural strength of MOSC and the bending–compressive strength ratio of hardened MOSC. MK and FA had an obvious toughening effect on MOSC, and the toughening effect of the compound addition of MK and FA on hardened MOSC was better than that of single-doped MK and single-doped FA [24].

### 3.3. Water Resistance

The water resistance of MOSC can be characterized by the softening coefficient; a higher softening coefficient denotes greater water resistance of MOSC. Figure 7 depicts the residual compressive strength of the MOSC samples after 28 days of water immersion, which all decreased compared to pre-submersion values. After 28 days of immersion in water, the residual strengths of samples C, C-FA, C-MK, and C-(MK+FA) were 75.59, 81.54, 88.68, and 96.19 MPa, respectively. The softening coefficients compared to sample C were 0.80, 0.84, 0.88, and 0.91, respectively. Furthermore, the strength loss values of C, C-FA, C-MK, and C-(MK+FA) were 20%, 15%, 12%, and 9%, respectively. These results show that the softening coefficients of the experimental groups (C-FA, C-MK, and C-(MK+FA)) were higher than that of control C, suggesting that the addition of admixtures can improve the water resistance of MOSC, and a synergistic enhancement can be observed with the compounded addition of MK and FA.

### 3.4. XRD Analysis

Analyzing the composition of the hydration products can aid in explaining the macroscopic properties of hardened MOSC. Figure 8 presents the XRD spectra of C, C-10FA, C-10MK, and C-(5MK+5FA). Table 3 summarizes the composition of hardened MOSC hydration products cured to 28 days quantitatively analyzed by software Topas4.2. C-10MK indicates that 10% of the MgO weight was replaced by MK; the other labels follow a similar naming convention. As shown in Figure 8, the main peaks observed in the MOSC samples cured to 28 days were the 517 phase, MgO, Mg(OH)_2_, MgCO_3_, SiO_2_, and Al_2_O_3_. The 517 phase represents complete MOSC hydration, while Mg(OH)_2_ indicates the generation of the incomplete hydration product; MgO, SiO_2_, and Al_2_O_3_ originated from lightly burned magnesia and mineral admixtures. MgCO_3_ primarily originated from the magnesite. In addition, the addition of mineral admixtures did not form new diffraction peaks, which suggests that the phase composition did not change in the hydration products. According to the quantitative results in Table 3, contrasting the control group C to C-10FA, C-10MK, and C-(5MK+5FA), a higher percentage of 517 phase can be observed in the non-control samples.

### 3.5. SEM Analysis

Figure 9 and Figure 10 present the SEM images of control group C, C-FA, C-MK, and C-(MK+FA) cured to 3 days and 28 days, respectively. As shown in Figure 9a, all the MOSC samples developed a small quantity of thin, needle-like crystals, which can be attributed to the 517 phase. However, in Figure 9b, the C-10FA samples exhibited an increased number of longer needle-like crystals than observed in sample C (Figure 9a). This observation confirms that addition of FA can promote the formation of 517 phases in MOSC. As displayed in Figure 9c, the presence of the 517 phase in C-10MK was significantly more apparent than that in C-10FA, indicating that MK generates more 517 phases than FA. Figure 9d exhibits a dense network of tightly wound, thick, and long 517 phase crystals formed in C-(5MK+5FA) after 3 days of curing. The extent of the 517 phase in the C-(5MK+5FA) sample was significantly greater than that in C-10MK and C-10FA. Therefore, the compounded effect of MK and FA on the system was superior in enlarging the strength-promoting phase to singly doped FA or MK.

According to Figure 9 and Figure 10, the samples cured to 28 days were similar to the samples cured to 3 days, although more phase 517 was detected in the system, which can be attributed to the effects of curing. According to Figure 10, it can be concluded that the single/double doping of MK and FA promoted the formation of phase 517 in the system. 

### 3.6. Mercury Intrusion Porosity (MIP)

Figure 11 and Table 4 present the pore characteristics of MOSC samples whether pristine or single-/double-doped with MK and FA at the optimum 10 wt.%. Figure 11a reveals the cumulative porosity of samples of C, C-FA, C-MK, and C-(MK+FA) after 28 days of curing. The porosity of the experimental samples was lower than that of the control group C; adding both MK and FA reduced the porosity of the system compared to adding only FA or MK. Figure 11b manifests the pore size distribution curve of the MOSC samples after 28 days of curing. The average pore size from smallest to largest started at C-(5MK+5FA) at 30.25 nm, followed by C-10MK, C-10FA, and control group C at 38.20 nm (Table 4). Compared to sample C, the average pore diameter of MOSC decreased by 20.81% when mixed with both MK and FA. Figure 12 illustrates the synergistic, filling effect of FA and MK on the MOSC samples, i.e., filling with particles of different sizes was more efficient in filling the void space than filling with MK or FA alone [25]. 

## 4. Discussion

As shown in Figure 2, the fluidity of C-FA slurry was better than that of C-(MK+FA), C, and C-MK. According to the Chinese National Standard GB/T51003-2014, when FA or MK was added to MOSC paste, in order to enable the composite system to reach the state of standard consistency, it was necessary to adjust the water consumption of the slurry. The water consumption of the FA and MK systems was 110% and 95%, respectively. According to the fixed water-to-binder ratio, due to different water requirements, the fluidity of MOSC paste differed as a function of the different mineral admixtures that were added, consistent with previous studies [26]. Since the particle size of MK was smaller than that of FA (Figure 1a), and the specific surface area of MK was significantly larger than that of FA (Figure 1b), more free water was required to wet the surface of MK, resulting in a greater fluidity of C-FA than C-MK. Due to the unique properties of MK and FA, the mobility curve of C-(MK+FA) was in between that of the individual components.

On the basis of Figure 3a, we concluded that FA would prolong the setting time of MOSC paste. These findings suggest that FA is capable of delaying the setting in C-FA, which agrees with previous studies [27]. The delayed setting time can be attributed to the dilution effect of FA. No new hydration products were generated in the C-FA gel (confirmed by Figure 8 and Table 3). Although the addition of FA is instrumental to establishing bridges between hydration products, FA also promotes the formation of phase 5 via the nucleation effect [28]. As shown in Figure 3b, MK shortened the setting time of MOSC. The rate of the hydration reaction largely affects the setting time of MOSC paste, and the rate of the hydration reaction is closely related to the concentration of magnesium sulfate solution in the system [29]. In general, a higher concentration of magnesium sulfate solution results in a faster hydration reaction rate and shorter coagulation time. With its larger specific surface area, MK can adsorb more free water. Therefore, the incorporation of MK reduces the fluidity of MOSC paste and increases the concentration of magnesium sulfate solution, thereby increasing the hydration reaction rate of MOSC and shortening the setting time [30]. Furthermore, the shortened setting may also be attributed to the synergistic effect of MK and FA on the MOSC paste. Although the addition of admixtures changed the setting properties of the cement paste, the setting time of pristine MOSC met both requirements of at least 45 min and less than 10 h for the initial and final setting times, respectively [31]. This shows that the addition of MK and FA allows MOSC to meet the requirements of national standards for the setting time of cement. Therefore, the setting time of MOSC paste can be controlled by adjusting the degree of FA and MK substitution in the cement compound. 

As shown in Figure 4 and Figure 5, the addition of MK and FA improved the mechanical properties of the hardened MOSC when the admixture was less than 10%, which can be attributed to the particle size distribution of the admixtures. Both FA and MK are good fillers [32]. Particles with smaller particle size could fill the pores of the cement system, making the structure of the hardened MOSC more compact and improving the mechanical properties. When the admixture content exceeded 10%, the mechanical properties of MOSC gradually deteriorated, which may be due to the fact that an excessive amount of admixture could no longer fill the pores, thereby reducing the strength. Furthermore, increasing the admixture content decreased the relative amount of α-MgO, which also decreased the amount of the 517 phase, thereby reducing the hydration reaction activity and overall strength [29]. The mechanical properties of C-FA, C-MK, and C-(MK+FA) were better than those of C, which was attributed to the filling and nucleation effects of FA and MK [33]. The particle sizes of FA and MK were much smaller than that of MgO, which facilitated the dispersion of FA and MK into the slurry and makes the MOSC slurry more uniform. The accumulation of spherical FA particles and MK particles filled the voids, allowing the particles and the matrix to accumulate more closely (confirmed by Figure 12). FA and MK particles as the precipitation core of Mg (OH)_2_ and 517 phases filled the gap between admixtures particles and MgO particles. FA and MK particles with smaller particle size reduced the pore size and provided more nucleation sites for hydration products. Therefore, when fine FA and MK particles were dispersed into the slurry, the nucleation effect accelerated the hydration reaction [34]. Moreover, the size gradation of the compound addition of FA and MK was better than that of MK or FA. In addition, compared with C, C-MK, and C-FA, the C-(MK+FA) sample produced more 517 phases (confirmed by Figure 8, Figure 9 and Figure 10 and Table 3). The 517 phases were primarily responsible for the strength of MOSC [35]. Therefore, to improve the mechanical properties of MOSC, the addition of mineral admixtures was beneficial to the cementitious system. However, considering the dilution effect and the strength reduction due to the decrease in hydration activity, both caused by an excessive amount of the admixture, the admixture content should be optimized according to the production needs.

As shown in Figure 6, the improvement effect of FA and MK on the flexural strength of the system was better than that on the compressive strength. This may be because the addition of FA and MK reduced the pore size and did not necessarily reduce the pore volume. Compared with the compressive strength, the flexural strength was more susceptible to the maximum particle size [36]. 

The porosity and pore size distribution of hardened MOSC also greatly influenced the mechanical properties. The MOSC pores can be divided into three distributions: gel pores (D < 10 nm), capillary pores (10 nm < D < 100 nm) and macropores (D > 100 nm) [37]. A greater number of macropores leads to greater negative effects on both the mechanical properties and the water resistance of MOSC. It can be seen from Figure 11 and Table 4 that the addition of FA and MK reduced the proportion of harmful macropores, which can be ascribed to MK and FA gradually filling the macropores. Regardless of single or multiple doping, the addition of FA and MK reduced the macropore content, increased the number of capillary pores, and reduced the overall porosity of the hardened MOSC, thus improving the mechanical properties and water resistance [25], as also confirmed by Figure 4 and Figure 5. In general, a greater density of the MOSC microstructure results in higher water resistance [32]. Therefore, water immersion did not erode C-(MK+FA) as much as the other samples, resulting in better water resistance than C, C-FA, and C-MK. The improvement in water resistance of C-FA, C-MK and C-(MK+FA) may be related to the formation of amorphous aluminum silicate gel by the reaction of active SiO_2_ and Al_2_O_3_ in FA or MK under the alkaline environment of MOSC system. These gels, when combined with phase 517, prevent further erosion of water [14].

## 5. Conclusions

The addition of FA to the MOSC paste extended the setting time and delayed the hydration process. However, adding MK to the system exhibited the opposite effects. When MK and FA were added together, the cement setting was accelerated. Therefore, the setting time of MOSC can be customized by adjusting the MK and FA content added into the system. Since the composition of the hydration products did not change with the addition of MK or FA, the admixtures merely contributed via the physical filling effect. The composite addition of MK and FA to MOSC can promote the system to produce more 517 phase, thereby improving the mechanical properties and water resistance.

## Figures and Tables

**Figure 1 materials-15-01334-f001:**
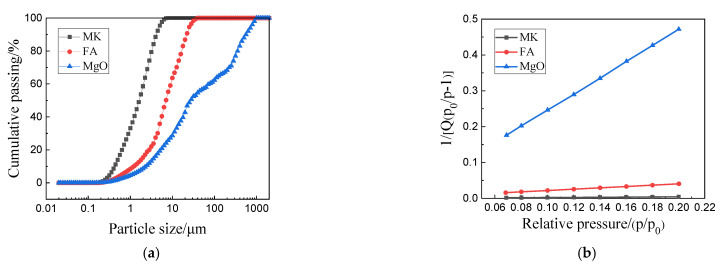
(**a**) Physical adsorption curves; (**b**) specific surface areas of MgO, FA, and MK.

**Figure 2 materials-15-01334-f002:**
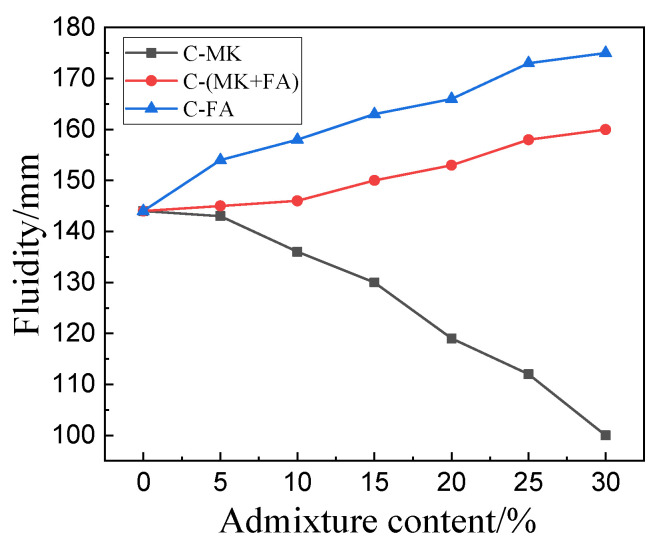
Fluidity of MOSC with different admixtures and admixture amounts.

**Figure 3 materials-15-01334-f003:**
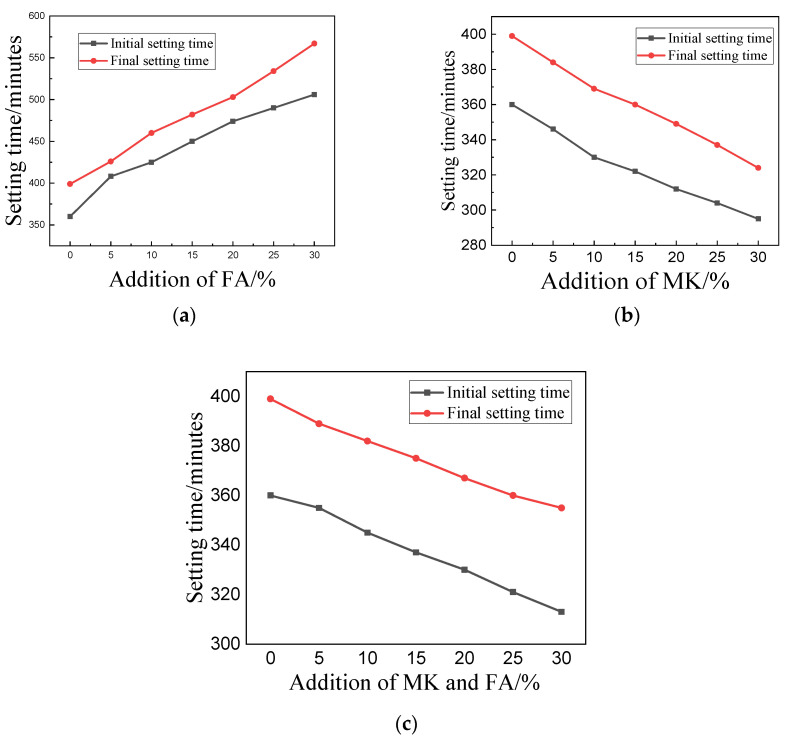
Setting times of MOSC composite pastes: (**a**) C-FA; (**b**) C-MK; (**c**) C-(MK+FA).

**Figure 4 materials-15-01334-f004:**
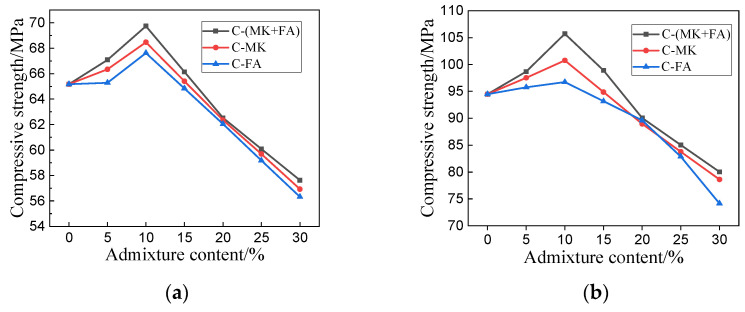
Compressive strength of hardened MOSC after different curing durations: (**a**) 3 days; (**b**) 28 days.

**Figure 5 materials-15-01334-f005:**
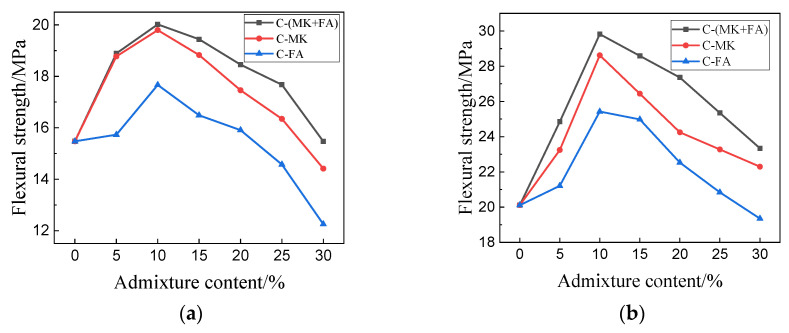
Flexural strength of hardened MOSC after different curing durations: (**a**) 3 days; (**b**) 28 days.

**Figure 6 materials-15-01334-f006:**
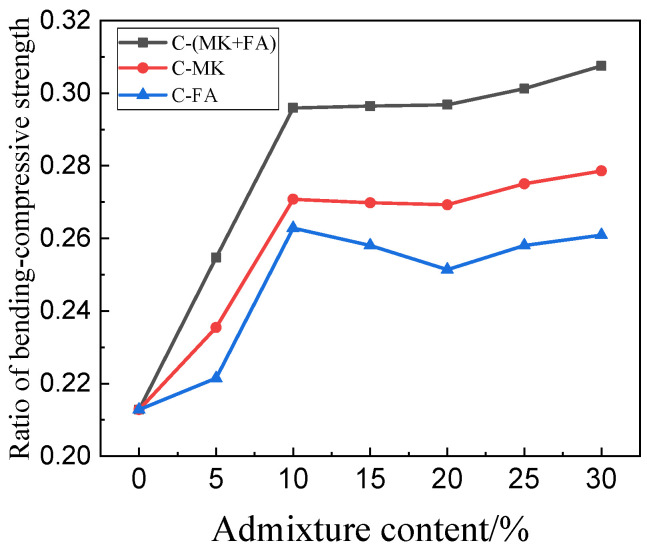
Ratio of bending–compressive strength of hardened MOSC after 28 days of curing.

**Figure 7 materials-15-01334-f007:**
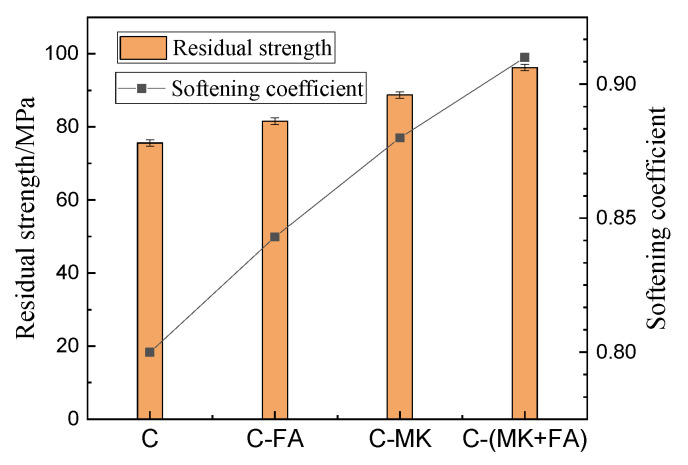
The residual strength and softening coefficient of MOSC with different admixtures immersed in water for 28 days (admixture content: 10%).

**Figure 8 materials-15-01334-f008:**
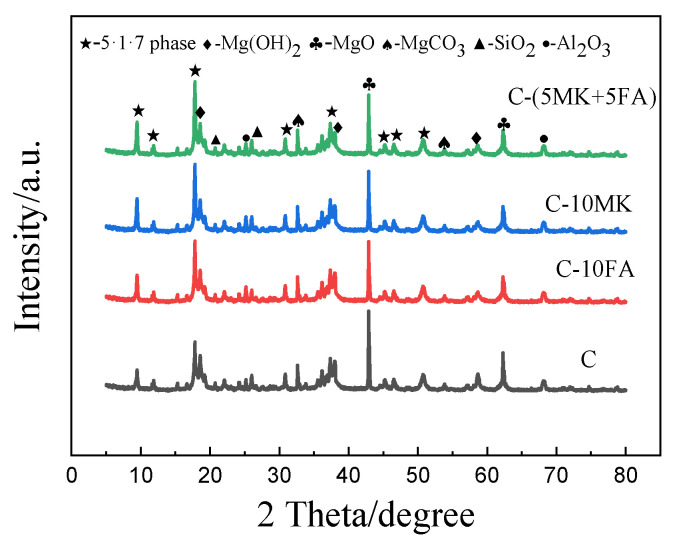
XRD patterns of MOSC with different mineral admixtures after 28 days of curing.

**Figure 9 materials-15-01334-f009:**
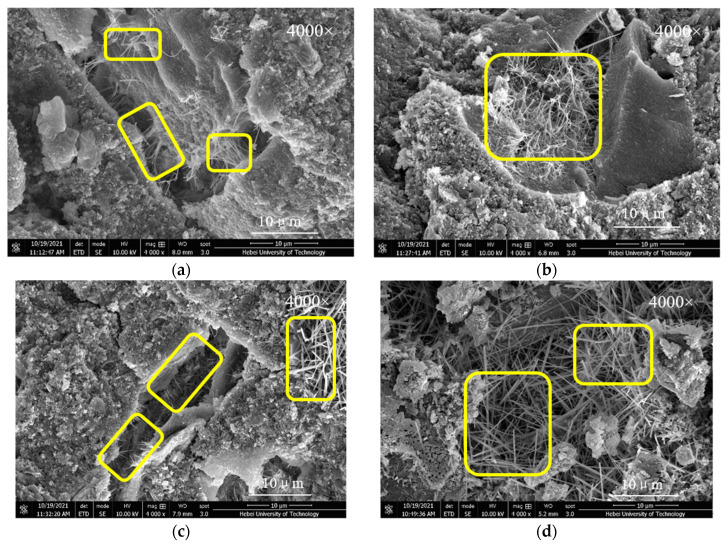
SEM images of MOSC with different mineral admixtures after 3 days of curing: (**a**) C; (**b**) C-10FA; (**c**) C-10MK; (**d**) C-(5MK+5FA). Remark: Phase 517 is shown in the yellow frame.

**Figure 10 materials-15-01334-f010:**
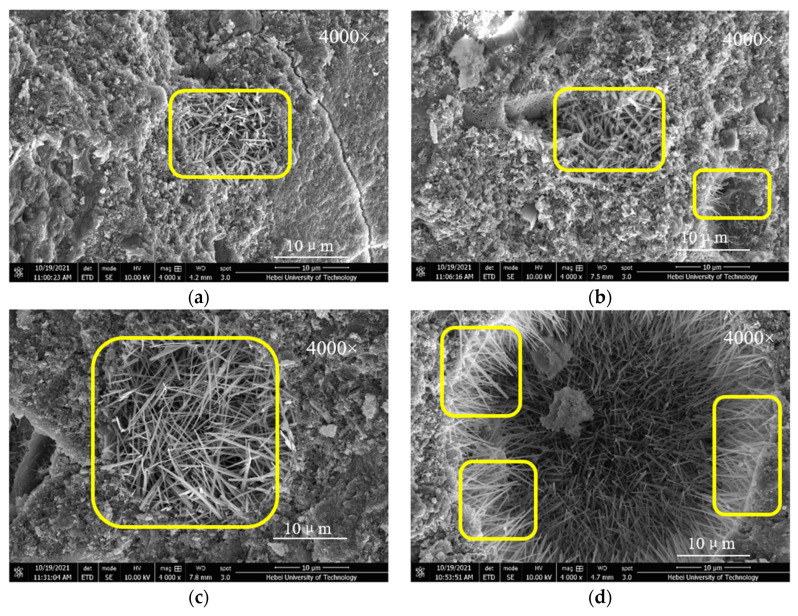
SEM images of MOSC with different mineral admixtures after 28 days of curing: (**a**) C; (**b**) C-10FA; (**c**) C-10MK; (**d**) C-(5MK+5FA). Remark: Phase 517 is shown in the yellow frame.

**Figure 11 materials-15-01334-f011:**
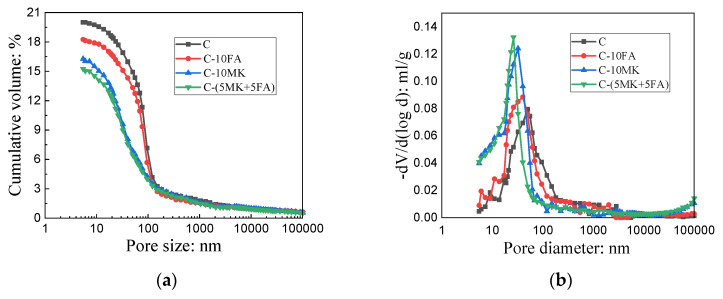
Pore characteristics of MOSC samples after 28 days of curing: (**a**) cumulative volume; (**b**) pore diameter distribution.

**Figure 12 materials-15-01334-f012:**
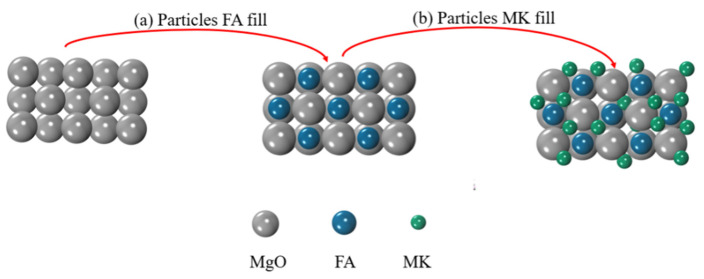
Scheme depicting the filling effect of FA and MK on MOSC samples: (**a**) filling with FA particles; (**b**) filling with MK particles.

**Table 1 materials-15-01334-t001:** Chemical composition of precursor materials: magnesia, FA, and MK (%).

Material	MgO	SiO_2_	CaO	Fe_2_O_3_	SO_3_	K_2_O	Al_2_O_3_	LOI	Others
Magnesia	83.34	6.99	2.13	0.80	0.13	0.03	2.38	2.21	1.99
FA	0.57	56.84	4.18	6.14	0.42	2.91	25.95	2.60	0.39
MK	0.02	58.05	0.29	0.85	0.10	0.23	38.15	0.40	1.91

**Table 2 materials-15-01334-t002:** The mix proportion of MOSC (g).

No		Magnesia	MgSO_4_·7H_2_O	FA	MK	Water
1	C	554.91	192.68	0	0	252.41
2	C-FA	527.15	192.68	27.76	0	252.41
3		499.41	192.68	55.50	0	252.41
4		471.67	192.68	83.24	0	252.41
5		443.93	192.68	110.98	0	252.41
6		416.19	192.68	138.72	0	252.41
7		388.43	192.68	166.48	0	252.41
8	C-MK	527.15	192.68	0	27.76	252.41
9		499.41	192.68	0	55.50	252.41
10		471.67	192.68	0	83.24	252.41
11		443.93	192.68	0	110.98	252.41
12		416.19	192.68	0	138.72	252.41
13		388.43	192.68	0	166.48	252.41
14	C-(MK+FA)	527.15	192.68	13.88	13.88	252.41
15		499.41	192.68	22.75	22.75	252.41
16		471.67	192.68	41.62	41.62	252.41
17		443.93	192.68	55.49	55.49	252.41
18		416.19	192.68	69.36	69.36	252.41
19		388.43	192.68	83.24	83.24	252.41

**Table 3 materials-15-01334-t003:** Phase composition of the MOSC after 28 days of curing analyzed using Topas4.2 (%).

Samples	517 Phase	Mg(OH)_2_	MgCO_3_	MgO	SiO_2_	Amor.	Rwp.
C	48.56	14.79	8.72	11.18	0.74	15.02	12.09
C-10MK	49.37	13.15	6.95	12.29	3.58	14.67	11.33
C-10FA	50.68	13.89	7.26	13.66	2.42	12.10	10.45
C-(5MK+5FA)	51.60	11.22	8.12	12.38	2.81	11.88	9.28

Amor.: amorphous phase. Rwp.: fitting error with Topas4.2.

**Table 4 materials-15-01334-t004:** Pore characteristics of MOSC at 28 days.

Sample	Porosity (%)	Average Pore Diameter (nm)	<10 nm (%)	10–100 nm (%)	>100 nm (%)
C	19.99	38.20	7.64	70.75	21.61
C-10FA	18.24	35.18	6.02	72.64	21.34
C-10MK	16.27	31.83	2.21	76.93	20.86
C-(5MK+5FA)	15.25	30.25	2.58	78.27	19.15

## Data Availability

The study did not report any data.

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
