# Peer review of "Effect of Fly Ash and Metakaolin on Properties and Microstructure of Magnesium Oxysulfate Cement"

_materials, 2022, doi:10.3390/ma15041334_

Round 1
Reviewer 1 Report
Article analysis the impact of fly ash and metakaolin admixtures on mechanical properties and microstructure of magnesium oxysulfate cement. The article is interesting but it needs to be revised:
1) I strongly suggest to increase the scaling of x and y axes in all Figures because the numerical values and titles can be barely seen.
2) How many samples were tested for every property? Any statistical deviations or upper and lower limits are not presented. No line connection between the results should be done because you do not know hat happens, e.g. at 15% admixture content in Figure 5. The connecting line now shows that at 15% admixture content, flexural strength reduces but authors did not test flexural strength at 15% admixture content, it may be higher than that at 10% admixture content. Please do additional testing or remove connecting line in graphs.
3) Please use another font colour in Figures 9-10, it is hard to see anything in those images. Additionally, I strongly suggest putting scaling bars in SEM images in the right upper corner so they could be seen together with their numerical values. Also, please indicate the magnification in the captions of Figures 9-10.
Author Response
Please see the attachment 1.

Reviewer 2 Report
The work presented for review is devoted to the improvement of the properties of magnesium oxysulfate cements, which is proposed to be done by adding fly ash and metakaolin to the composition of the cement system.
General remarks:
It should be noted that the work does not contain the Discussion part. At the same time, the discussion of separate results often "gets ahead of itself" and contains links to Figures and information presented later. This may be found, for example, when discussing Figure 2 (lines 180-187); when discussing Figure 4 (lines 245-247), when discussing Figure 7 (lines 286-287) etc.
In my opinion, the authors should reduce the Results section, limiting them to only those facts which the relevant sections are devoted, and try to analyze the general patterns in the newly created Discussion section. This remark is the main one for sending the article to a Major revision.
After preparing the Discussion section, the authors should also change the form of submission of the Conclusions section, eliminating the numbering of paragraphs, reducing the conclusions to one or two generalizations of the results obtained.
A note on research methods:
1. The phase card numbers are not given for the RFA method and there are no references to the corresponding databases used (for example, ICDD PDF2).
Design notes:
1. Figures 2, 3, 4, 5, 6 does not contains errors in determining the corresponding properties, the same applies to Tables 3 and 4. The presentation of data in these tables should be given taking into account the errors. The lines connecting the points in these figures should be dot-lined, for better visual presentation.
2. Figure 7, in addition to the absence of error bars, contains bars with a solid filling. The hatching will look better.
3. The text (signature of phase 5.1.7) on the microphotographs of Figures 9 and 10 is poorly readable. In addition, the arrows leading to the text pointer interfere with the perception of the drawing. It is necessary to leave the frames of the selected zones, removing the arrows and the signature in the figures, and add the captions with a comment that the areas of phase 5.1.7 are highlighted with frames.
Noticed typos and so on:
Line 93: a (.) is missing
Line 212: (C) should be (c)
Line 261: "maximum size" of what? Pores, particles?
Line 288: In general (not "Generally speaking")
Author Response
Please see the attachment 1.

Round 2
Reviewer 1 Report
Authors have corrected the article according to my remarks.
Reviewer 2 Report
Thank you for taking into account the remarks, the article in my opinion can be published in its current form